# Trend Detection of Wave Parameters along the Italian Seas

**Tommaso Caloiero [1] and Francesco Aristodemo [2,***

[1] National Research Council—Institute for Agricultural and Forest Systems in Mediterranean (CNR-ISAFOM), 87036 Rende, Italy; tommaso.caloiero@cnr.it

[2] Department of Civil Engineering (DINCI), University of Calabria, 87036 Rende, Italy

* Correspondence: francesco.aristodemo@unical.it; Tel.: +39-09-8449-6554

**Abstract:** In this paper, trend detection of wave parameters such as significant wave height, energy period, and wave power along the Italian seas was carried out. To this purpose, wave time series in the period 1979–2018 taken from the global atmospheric reanalysis ERA-Interim by European Center for Medium-Range Weather Forecasts (ECMWF) were considered. Choosing a significance level equal to 90%, the use of the Mann–Kendall test allowed estimating ongoing trends on the mean values evaluated at yearly and seasonal scale. Furthermore, the assessment of the magnitude of the increase/decrease of the wave parameters was performed through the Theil–Sen estimator. The obtained results underlined that the mean values of the considered wave parameters were characterized by a high occurrence of positive trends in the different Italian seas. The findings of this study could have implications for studies of coastal flooding, shoreline variations, and port operations, and for the assessment of the performances of Wave Energy Converters.

**Keywords:** Italian Seas; trend detection; significant wave height; energy period; wave power

## 1. Introduction

The Italian seas are usually characterized by a high number of offshore and coastal activities with increased exploitation of various types of marine resources (e.g., [1]). Furthermore, as highlighted by the fifth report of the International Panel on Climate Change [2], the Mediterranean, including all the Italian seas, represents one of the most exposed zones in the world to the impacts of global warming. The high variability of several climatic variables render the Mediterranean basin a hotspot for climate change processes (e.g., [3]). Among the hazards exacerbated by climate change, wind-generated waves appear to be among the most significant. Indeed, their importance arises from different studies and applications occurring offshore such as navigation safety, ship routes, oil and gas production and transportation, respectively, derived by the use of platforms and pipelines. Furthermore, important implications also arise from nearshore processes such as port operations, coastal vulnerability assessment, and coastal structures design and verification. On the other hand, the potential increase/decrease of wave parameters is a key factor in the change of coastal flooding and erosion processes [4,5], and of the real performances of devices adopted to transform wave energy into electricity (e.g., [6,7]).

To study possible trends of wave parameters, it was observed that more than 30 years of wave series are necessary [8]. Wave climate variability and trends were broadly analyzed in different seas of the world by the application of various datasets (e.g., [9–12]). Paying attention to the Italian seas, some studies were carried out. In particular, Musić and Nicković [13] analyzed trends at annual scale of the 50th, 90th, and 99th percentile of the significant wave height, Hs, along the Ionian Sea. Martucci et al. [14] performed a trend analysis of average and extreme values of $H_s$ for some nodes belonging to the Italian seas in the period 1958–1999, underlining the occurrence of an initial negative trend in $H_s$ and a subsequent positive trend. Young et al. [15] performed a modelling of the yearly mean of the 90th and 99th percentile of $H_s$, obtaining an overall positive trend in the time

window 1985–2008 along the Italian seas. In a specific point belonging to the Adriatic Sea, Pomaro et al. [16] analyzed $H_s$ trends in the period 1979–2015, observing a decrease in the highest percentile (99th) and a rise of the 50th and 75th ones. With reference to the Calabria region in southern Italy, Caloiero et al. [17] deduced slightly positive trends in the yearly and seasonal mean values of $H_s$ and relevant positive trends in those linked to the energy period $T_e$. More recently, a trend detection over the whole Mediterranean Sea for maxima, 98th percentile and mean $H_s$ at yearly scale was performed by De Leo et al. [18], highlighting different trends comparing mean and maximum values.

The aim of this study is to detect and quantify the occurrence of trends in the significant wave height along the Italian seas, also including, for the first time, wave parameters such as energy period and wave power, $P$. To the best of the authors' knowledge, possible trends linked to these two last parameters were only studied by Ulazia et al. [19] in some points belonging to the Bay of Biscay. Within this context, this study aims to analyze the trends of $P$ and the wave parameters, $H_s$ and $T_e$, included in its calculation. In particular, considering seasonal and yearly scale, the mean values of $H_s$, $T_e$, and $P$ were investigated. A preliminary assessment was performed by Caloiero et al. [20]. Besides the analyzed variables, another added value of this work is the adopted database, i.e., ERA-Interim by ECMWF, which was not previously used in research related to the trend detection of wave parameters. In fact, although some studies on the trend behavior were performed along the Italian seas [13–16], these analyses were concentrated in some specific locations and using old databases. Only De Leo et al. [18] performed a trend analysis of the mean annual values of the significant wave height in the Mediterranean Sea, but using a wave dataset characterized by a different space-time resolution.

The contents of the present work are organized as follows. First, the study area and the adopted wave data are briefly illustrated. Then, the estimate of the involved wave parameters are described as well as the methodologies to carry out the trend analysis. The following section describes the trend detection on the mean wave parameters at annual and seasonal scale, including a critical discussion of the results compared to those of previous studies. Finally, the conclusions close the paper.

## 2. Study Area and Wave Data

Italy is a country consisting of a continental part, delimited by the Alps, a peninsula and several islands surrounding it such as Sicily and Sardinia. It is placed between latitudes 35° and 47° N, and longitudes 6° and 19° E. This country has a total coastline length of about 7600 km including all the islands. The Italian seas are located within the Mediterranean Sea. Specifically, four seas surround the Italian Peninsula: the Adriatic Sea in the east, the Ionian Sea in the south, and the Ligurian Sea and the Tyrrhenian Sea in the west. The greatest water depth of the Italian seas is more than 5000 m and located in the Ionian Sea.

Here, the wave parameters $H_s$ and $T_e$ were deduced from the global atmospheric reanalysis ERA-Interim by the European Center for Medium-Range Weather Forecasts (ECMWF). The considered time window refers to the period 1979–2018, while the spatial resolution of the wave nodes is 0.75° × 0.75° and the time sampling between two successive wave data is 6 h. Figure 1 shows the used 129 wave nodes along the Italian seas and the name of the main marginal seas. The adopted ECMWF grid points are located at water depths $d$ ranging from 17.7 to 4098.3 m.

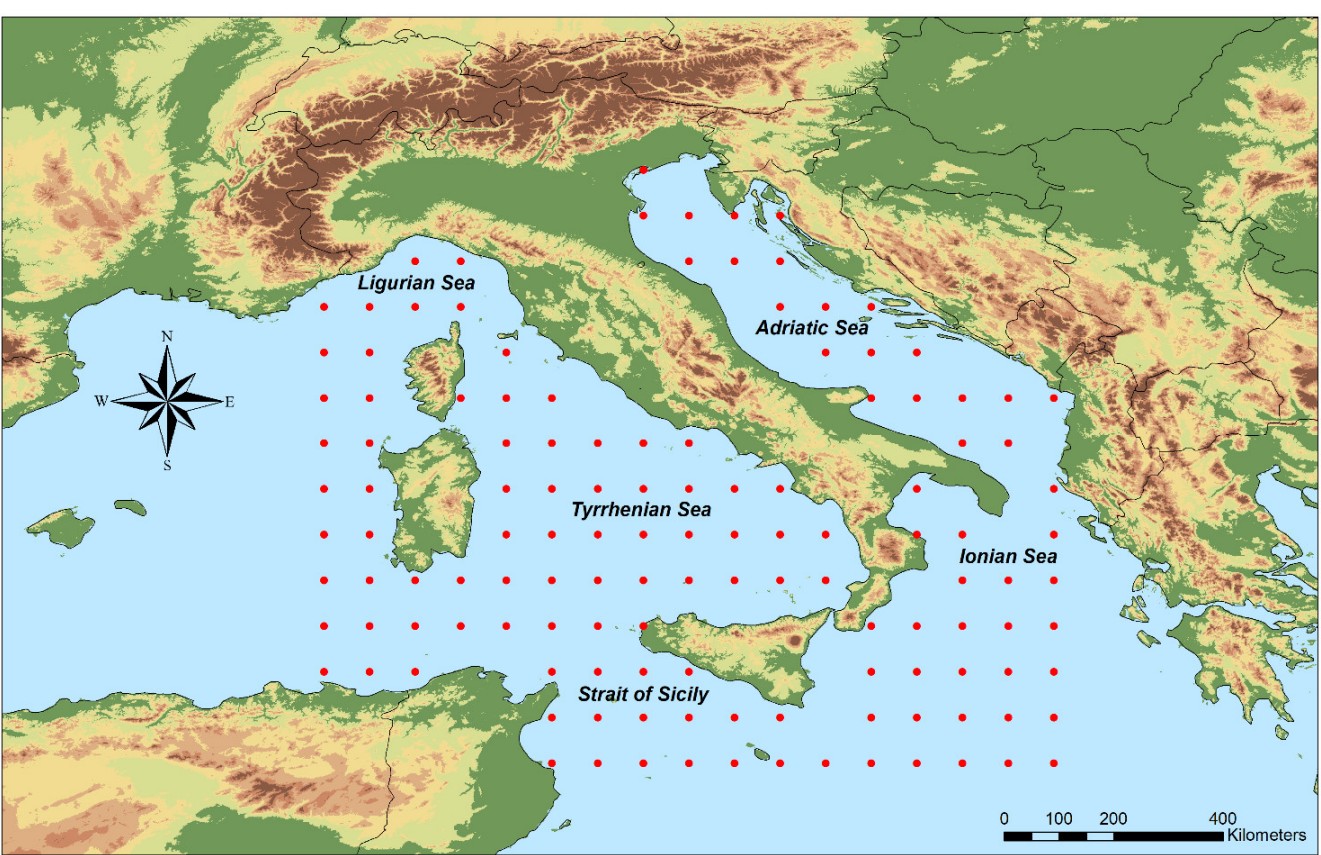

**Figure 1.** Target sea area and calculation of ECMWF grid points.

## 3. Methodology

The synthetic parameters of sea states adopted in the current study are $H_s$ and $T_e$. Furthermore, the wave power per unit crest length, $P$, is also considered.

Adopting the spectral approach, $H_s$ is due to:

$$H_s = 4\sqrt{m_0}, \tag{1}$$

where $m_0$ is the zero-th spectral moment given by:

$$m_0 = \sum_{ij} S_{ij} \Delta f_i \Delta \theta_j, \tag{2}$$

where $S_{ij}$ is the wave power density function over the $i$-th frequency and $j$-th direction, $\Delta f_i$ is the $i$-th frequency width of the density and $\Delta \theta_j$ is the $j$-th angular width of the density.

The value of $T_e$, representing the variance-weighted mean period of the one-dimensional density spectrum, is evaluated as follows:

$$T_e = \frac{m_{-1}}{m_0}, \tag{3}$$

where $m_{-1}$ is the minus-one spectral moment due to:

$$m_{-1} = \sum_{ij} f_i^{-1} S_{ij} \Delta f_i \Delta \theta_j, \tag{4}$$

where $f_i$ is the $i$-th frequency.

The value of $P$ is generally given by:

$$P = E c_g, \tag{5}$$

being $E$ the wave energy calculated as:

$$E = \frac{\rho g}{16} H_s^2, \tag{6}$$

being $\rho$ (water density) $\approx 1025$ kg/m$^3$ and $g$ (gravitational acceleration) $\approx 9.806$ m/s$^2$.

In Equation (5), $c_g$ represents the group celerity, which is determined as:

$$c_g = \frac{c}{2}\left[1 + \frac{2kd}{sinh(2kd)}\right],$$  (7)

being $d$ the depth of the water, $c$ (mean celerity) = $L_m/T_e$, $k$ (mean wave number) = $2\pi/L_m$, and $L_m$ is the mean wave length determined by the canonical linear dispersion equation:

$$L_m = \frac{gT_e^2}{2\pi}tanh(kd),$$  (8)

The wave parameters $H_s$ and $T_e$ were preliminarily processed to find possible inconsistencies to be eliminated for the subsequent trend analysis. In particular, the following criteria were used to remove unreliable data (e.g., [21]):

−    occurrence of zero and repeated data;
−    outlier values;
−    $|H_{s,i+1} − H_{s,i}| > 1.5$ m with $|\theta_{i+1} − \theta_i| < 30°$, being $\theta$ the mean wave direction;
−    $T_{p,i}/T_{e,i} > 2$, where $T_p$ is the peak period;
−    $H_{s,i}/L_{p,i} < 0.1$.

where $L_p$ (peak wave length) = $gT_p^2$ tanh$(kd)/(2\pi)$.

The mean efficiency due to the ratio between filtered data and rough ones is very high and equal to 99.94% involving all wave nodes.

The possible presence of temporal tendencies in the $H_s$, $T_e$, and $P$ series has been assessed with two well-known non-parametric tests. In particular, the statistical significance was assessed with the Mann–Kendall (MK) non-parametric test [22,23] for a significance level equal to 90%. The slopes of the trends were instead calculated by the Theil–Sen that is generally considered more powerful than linear regression methods in trend magnitude evaluation, because it is not subject to the influence of extreme values [24].

## 4. Results and Discussion

Before analyzing the possible trends obtained through the application of the Mann–Kendall test with their quantification due to the Theil–Sen estimator, an assessment of mean wave climate along the Italian seas at yearly scale is briefly illustrated. By analyzing the studied wave parameters, the greatest values of $H_s$, i.e., >1 m were identified off the western area of Corsica and Sardinia, between Sicily and Sardinia and off the southern part of Sicily. Conversely, the lowest $H_s$ were observed along the northern zones of Adriatic and Ionian Seas with values lower than 0.4 m. Considering $T_e$, the largest values greater than 4.5 s were found in the central-southern part of the Italian seas. Yet, the lowest $T_e$ were observed in the northern part of the Adriatic basin. Owing to its quadratic dependency on $H_s$, the greatest values of $P$, i.e., >6 kW/m, were detected in similar nodes of those observed for $H_s$ and, specifically, off the western zone of Corsica and Sardinia and between Sicily and Sardinia. The lowest $P$ have been substantially identified near the coasts belonging to the Ligurian, Tyrrhenian, Ionian, and Adriatic Seas with values lower than 3 kW/m. The above findings generally agree with those determined by Lionello et al. [25] and Besio et al. [26] who adopted different wave datasets.

The results obtained through the trend analysis related to the yearly and seasonal mean $H_s$ values are shown in Figures 2 and 3.

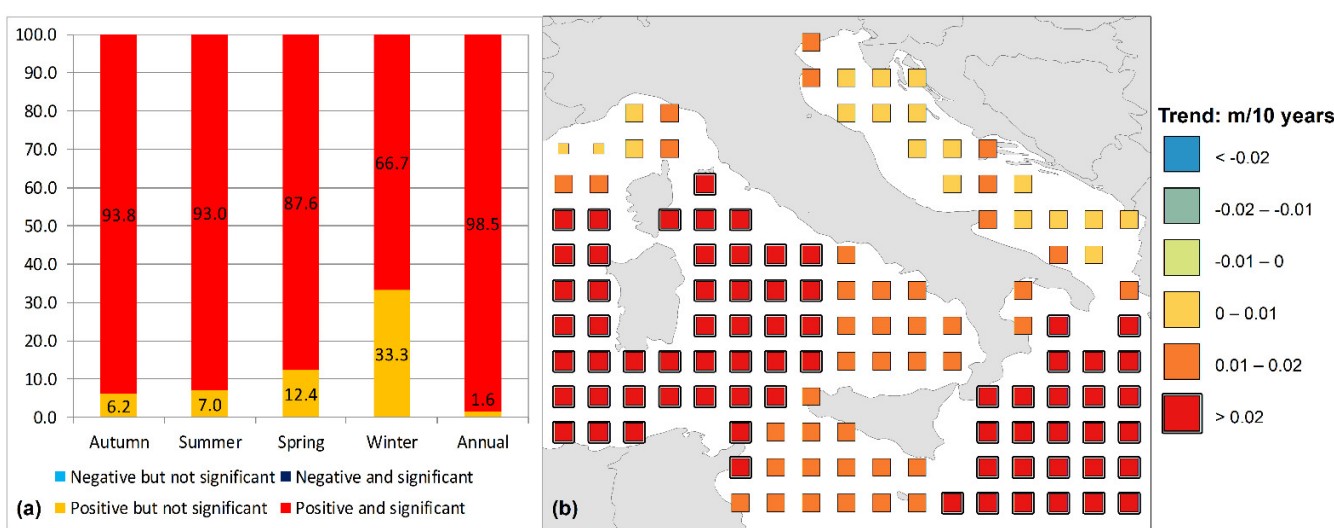

**Figure 2.** (**a**) Seasonal and yearly percentages of grid points presenting positive or negative trends in the mean $H_s$ values and (**b**) spatial distribution of the yearly trend of mean $H_s$. Large dimension squares reveal a significant trend, while small squares show a not significant trend.

**Figure 3.** Spatial distribution of the seasonal trend of the mean $H_s$ values for (**a**) winter, (**b**) spring, (**c**) summer, and (**d**) autumn. Large dimension squares reveal a significant trends, while small squares show a not significant trend.

Specifically, Figure 2a highlights the percentages of grid points with a positive or negative trend for $H_s$. In this case, only positive trends were identified at yearly scale.

Indeed, about 98.4% of the grid points showed a significant positive trend. As highlighted in Figure 2b, positive trends of annual mean $H_s$ values occur for all Italian seas, where the largest positive ones were detected in the areas highlighted by a thicker square with values larger than 0.02 m/10 years. These areas include the western zones of Corsica and Sardinia, the western part of the Tyrrhenian Sea and the Ionian Sea. In particular, the highest value is 0.046 m/10 years and identified near the Algerian coast. Note that in this Figure and in the subsequent Figures containing the spatial distribution of the trends, squares dimension suggests the significance level (SL) of the trend. Specifically, large dimension squares reveal significant trend (SL = 90%), while small squares otherwise, i.e., a not significant trend. Positive trends of annual mean $H_s$ values ranging from 0.01 m to 0.02 m/10 years were mainly detected in the eastern area of the Tyrrhenian Sea and near the Strait of Sicily. The Adriatic and Ligurian Seas represent the areas substantially characterized by positive trends of yearly mean $H_s$ values with the smallest magnitude, i.e., oscillating between 0 and 0.01 m/10 years.

Paying attention to the trend detection carried out at seasonal scale regarding the average $H_s$ values, Figure 2a describes the occurrence of only positive trends during all seasons. In winter, about 67% of the wave data highlighted significant positive trends. During spring, the percentage of significant positive trends is higher than the previous season, and approximately equal to 88% of the grid points. The above percentage increases during summer and autumn, respectively, involving about 93 and 94% of the wave data.

Figure 3 describes the spatial distribution of the seasonal trend of the mean $H_s$ values along the Italian seas. The seasonal distribution was organized intro groups of months as follows: Winter (December, January, and February), Spring (March, April, and May), Summer (June, July, and August), and Autumn (September, October, and November).

In general, these results confirm what happens on a yearly scale with an occurrence of only positive trends, although with some differences in magnitude and location. During winter (Figure 3a), significant positive trend values of $H_s$ with increases of more than 0.02 m/10 years were detected in the areas characterized by a thicker square, including the western zone of the Tyrrhenian Sea, the western coast of Corsica and Sardinia, the Strait of Sicily, and the majority of the Ionian basin. The peak was identified between Sardinia and Algeria with a value equal to 0.056 m/10 years. Positive values of $H_s$ in the range 0.01–0.02 m/10 years were instead identified in the central-eastern area of the Tyrrhenian basin and in some parts belonging to the Adriatic Sea (Figure 3a). Some scattered wave nodes along the Tyrrhenian, Adriatic, and Ligurian Seas were finally characterized by values of $H_s$ oscillating between 0 and 0.01 m/10 years (Figure 3a). In spring (Figure 3b), the areas with positive trends for $H_s$ with an increase greater than 0.02 m/10 years were underlined by a thicker square and proved to be similar to those observed in winter, with the exception of larger zones with this projection related to the Tyrrhenian and Ionian Seas. The maximum value in this season was identified near the Strait of Sicily with a value of 0.038 m/10 years. Positive trends for $H_s$ oscillating between 0.01 and 0.02 m/10 years were detected in the southern parts of the Tyrrhenian and Adriatic Seas and in the northern part of the Ionian Sea. Some nodes along the Adriatic Sea were characterized by positive trends for $H_s$ ranging from 0 to 0.01 m/10 years. A similar behavior for $H_s$ was also identified in summer (Figure 3c). In this season, the differences compared to the previous seasons referred to the occurrence of positive trends with a small magnitude near the Strait of Sicily, of positive trends oscillating between 0.01 and 0.02 m/10 years in the Ligurian Sea and of positive trends ranging from 0 and 0.01 m/10 years in the whole Adriatic Sea (Figure 3c). The peak value equal to 0.044 m/10 years was detected in the Ionian Sea. As highlighted by a thicker square, autumn represents the season with the highest occurrence of positive trends for $H_s$ larger than 0.2 m/10 years along the Italian seas, with a maximum value of 0.049 m/10 years located between Sardinia and Algeria (Figure 3d). The Adriatic Sea is the basin with the occurrence of positive trends with the smallest magnitude.

The trend detection was also carried out for the annual and seasonal mean $T_e$ values (Figures 4 and 5). Differently from $H_s$, all wave nodes showed a significant positive trend both at yearly scale and at seasonal scale.

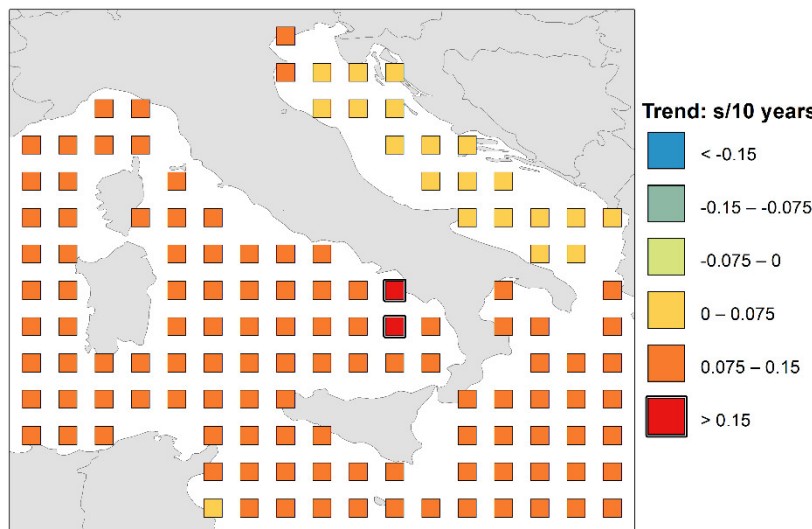

**Figure 4.** Spatial distribution of the yearly trend of mean $T_e$. Large dimension squares reveal a significant trend, while small squares show a not significant trend.

**Figure 5.** Spatial distribution of the seasonal trend of the mean $T_e$ values for (**a**) winter, (**b**) spring, (**c**) summer, and (**d**) autumn. Large dimension squares reveal a significant trend, while small squares show a not significant trend.

Figure 4 highlights the spatial distribution of the trends for the annual mean $T_e$ values along the Italian seas. Only significant positive trends having various magnitudes occur in the whole analyzed area. Positive trends of $T_e$ larger than 0.15 s/10 years were highlighted by a thicker square and concentrated in the eastern zone of the Tyrrhenian Sea with a peak value equal to 0.16 s/10 years. Positive trends of $T_e$ in the range 0.075–0.15 s/10 years were detected in the majority of the Italian seas and, specifically, along the Ligurian and Ionian Seas, the Strait of Sicily and in the central-western area of the Tyrrhenian Sea. The greatest increment in this range was detected near the Algerian coast. Positive trends of $T_e$ oscillating between 0 and 0.075 s/10 years were mainly observed along the Adriatic Sea.

Figure 5 illustrates the spatial variation of the seasonal trend analysis on the mean $T_e$ values in the Italian seas where only significant positive trends occur as in the annual analysis.

In winter (Figure 5a), positive trends of $T_e$ greater than 0.15 s/10 years were highlighted by a thicker square. They were detected along the Algerian coast, with a maximum value equal to 0.17 s/10 years, and for just one node in the Ionian Sea near the Calabrian coast. As observed during the mean year, positive trends of $T_e$ oscillating between 0.075 and 0.15 s/10 years cover almost the entire investigated area. Positive trends ranging from 0 to 0.075 s/10 years were instead individuated in some areas belonging to the Ionian, Ligurian, and Adriatic Seas. The spring season was characterized by one grid point having a positive trend of $T_e$ larger than 0.15 s/10 years and placed in the northern zone of the Tyrrhenian Sea (Figure 5b). As in winter and during the year, positive trends of $T_e$ in the range 0.075–0.15 s/10 years were observed in the Ligurian, Ionian, and Tyrrhenian Seas and in the Strait of Sicily. Positive trends of $T_e$ with the smallest magnitude were detected in some parts of the Ionian Sea and in the whole Adriatic basin. During summer (Figure 5c), positive trends of $T_e$ greater than 0.15 s/10 years were underlined by a thicker square and concentrated in the eastern area of the Tyrrhenian Sea, as appearing during the year, with a maximum value equal to 0.17 s/10 years. Concerning the positive trends of $T_e$ in the range 0.075–0.15 s/10 years, their occurrence is substantially the same of that individuated during the annual analysis. Positive trends of $T_e$ ranging from 0 to 0.075 s/10 years were detected in the northern part of the Ionian Sea and along the Adriatic and Ligurian Seas. Regarding autumn, positive trends of $T_e$ larger than 0.15 s/10 years were highlighted through a ticker square and mainly detected in the eastern part of the Tyrrhenian Sea, as occurring during the year and the summer, with a peak value of 0.17 s/10 years (Figure 5d). For the positive trends of $T_e$ in the range 0.075–0.15 s/10 years, the same spatial concentration occurred as for the previous seasons and during the year. Positive trends of $T_e$ in the range 0–0.075 s/10 years were concentrated in the Adriatic Sea.

The results obtained by the trend detection carried out on the annual and seasonal mean $P$ values are described in Figures 6 and 7. Specifically, Figure 6a highlights the percentages of wave data with a positive or negative trend involving $P$. As previously observed for the wave parameters, $H_s$ and $T_e$, which are linked to the determination of $P$ (see Equation (5)), the percentage of significant positive trends is high. Indeed, more than 90% of the wave data showed a significant positive trend at yearly scale.

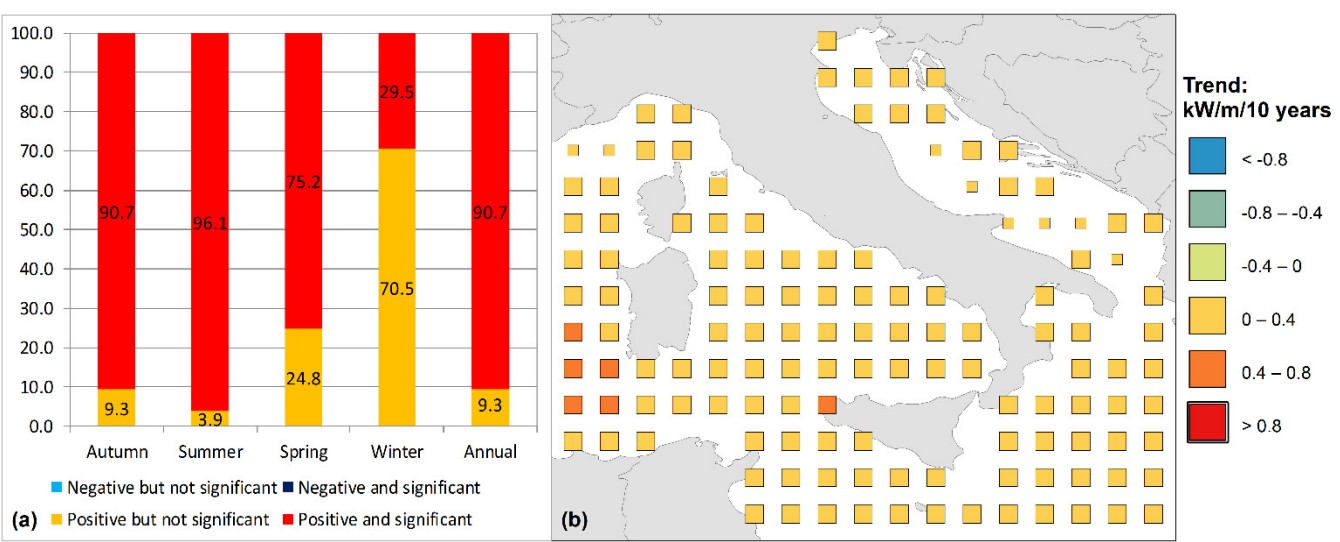

**Figure 6.** (**a**) Seasonal and yearly percentages of grid points presenting positive or negative trends in the mean *P* values and (**b**) spatial distribution of the yearly trend of mean *P*. Large dimension squares reveal a significant trend, while small squares show a not significant trend.

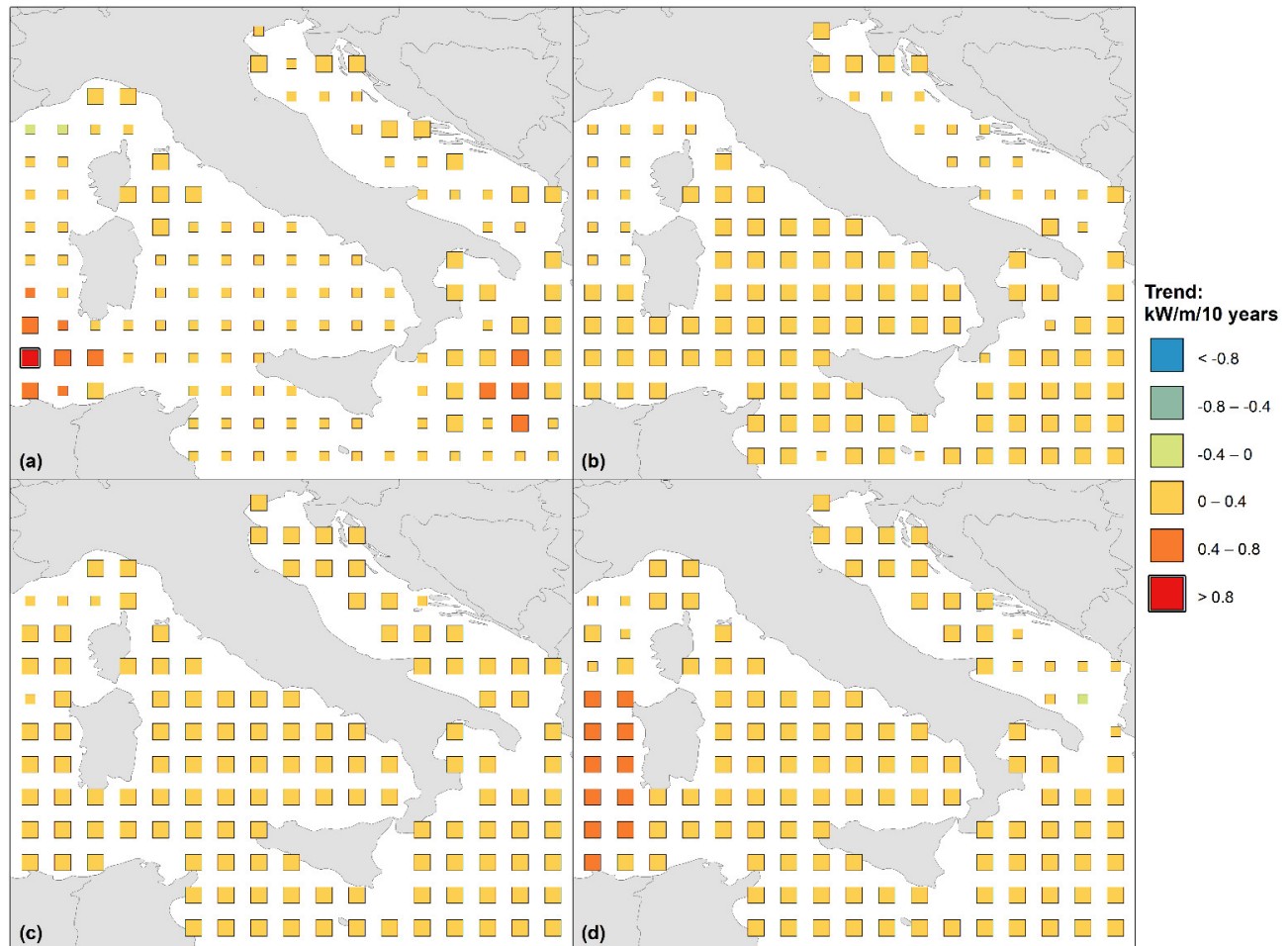

**Figure 7.** Spatial distribution of the seasonal trend of the mean *P* values for (**a**) winter, (**b**) spring, (**c**) summer, and (**d**) autumn. Large dimension squares reveal a significant trend, while small squares show a not significant trend.

Figure 6b illustrates the spatial distribution of the trends for the annual mean *P* values along the Italian seas. It can be observed that the major part of the involved grid points

showed a positive trend with values oscillating between 0 and 0.4 kW/m/10 years. As underlined by the use of a thicker square, just four wave nodes located between Sardinia and Algeria presented a positive trend with values ranging from 0.4 to 0.8 kW/m/10 years, where the peak value is approximately equal to 0.5 kW/m/10 years.

When the seasonal analysis is considered, the percentage of wave nodes showing a significant trend is quite variable (Figure 6a). In fact, significant positive trends associated to winter, spring, summer, and autumn appeared, respectively, in 29.5, 75.2, 96.1, and 90.7% of the grid points. While the winter season is characterized by a large percentage of positive trends but not significant, summer and autumn are distinguished by a percentage of positive trends but not significant, which is comparable with that observed during the year.

Figure 7 describes the spatial variation of the trends for the seasonal mean $P$ values along the Italian seas. During winter (Figure 7a), only one node highlighted by a thicker square and located close to the Algerian coast was characterized by the highest positive trend, which is larger than 0.8 kW/m/10 years and specifically equal to 0.86 kW/m/10 years. Positive trends for $P$ ranging from 0.4 to 0.8 kW/m/10 years were identified between Sardinia and Algeria and in the central zone of the Ionian Sea. Furthermore, positive trends for $P$ oscillating between 0 and 0.4 kW/m/10 years were linked to some grid points belonging to the Ionian, Adriatic, Tyrrhenian, and Ligurian Seas. In spring (Figure 7b), a large number of positive trends for $P$ in the range 0–0.4 kW/m/10 years occurred. These trends were detected in the Tyrrhenian and Ionian Seas, near Sicily and Sardinia and in the northern zone of the Adriatic basin. The maximum value is equal to 0.36 kW/m/10 years and placed between Sardinia and Algeria. A similar behavior appears during summer (Figure 7c). In fact, almost all grid points highlighted positive trends for $P$ ranging from 0 to 0.4 kW/m/10 years along the Italian seas with a maximum magnitude equal to 0.24 kW/m/10 years corresponding to a wave node located in the Ionian Sea. The last season, i.e., autumn, is characterized by a certain number of positive trends ranging from 0 to 0.8 kW/m/10 years (Figure 7d). In particular, positive trends oscillating between 0.4 and 0.8 kW/m/10 years appeared off the western part of Sardinia and near the Algerian coast with a peak value equal to 0.63 kW/m/10 years. Positive trends ranging from 0 to 0.4 kW/m/10 years were identified in the other areas belonging to the Tyrrhenian, Adriatic, Ligurian, and Ionian Seas. The maximum magnitude of the positive trend in the above range is equal to 0.39 kW/m/10 years and placed in the Ionian Sea near the Calabrian coast.

The local wind patterns and the magnitude of the marginal Seas, i.e., the extension of the fetches, may explain the trends at annual and seasonal scale. Indeed, the magnitude of the wave quantities generally agrees with the corresponding trends of this quantity at annual and seasonal scale, particularly for $H_s$ and $P$. For these two wave parameters, relevant trends refer to the areas, i.e., western zones of Corsica and Sardinia, western Tyrrhenian Sea and Ionian Sea, where the fetches are particularly significant. The highest trends refer to the western areas of Corsica and Sardinia, which are subject to strong Mistral and Tramontane winds coming from southern France and the northwestern Mediterranean Sea, particularly from the Gulf of Lion (e.g., [27,28]). The trends with the lower positive magnitude or not significant are substantially linked to the Adriatic and Ligurian Seas, where these semi-enclosed basins are characterized by little wind fields, i.e., Libeccio and Sirocco, in comparison with the other marginal seas, even though significant local winds such as Bora occur in limited areas [27].

Here, a critical discussion of the obtained results on the trend behavior of the involved wave parameters along the Italian seas compared to previous literature studies is performed. Considering only the Ionian Sea, contrary to the present results, Musić and Nicković [13] observed negative trends in the 50th percentile of $H_s$ adopting the wave model WAM forced by the atmospheric model REMO in the time window 1958–2001. Instead, the current findings agree with those provided by Martucci et al. [14] who detected the time evolution of $H_s$ on the basis of the WAM forced by the ERA-40 dataset in the time interval 1958–1999. Specifically, Martucci et al. [14] highlighted an increasing trend for the annual mean $H_s$ series starting from 1989. The present results also generally agree with

the analyses by Pomaro et al. [16], who underlined an increase in the 50th percentile of $H_s$ in the period 1979–2015 even though concentrated in a specific recording station in the northern zone of the Adriatic Sea. Using satellite measurements performed in the time window 1985–2008, Young et al. [15] detected positive trends on mean $H_s$ series along the Italian seas, with the exclusion of some small areas linked to the Ionian and Ligurian Seas. The obtained positive trends on mean $H_s$ values substantially agree with those detected by De Leo et al. [18], especially for the Ionian Sea, while discrepancies were observed in the Ligurian and Adriatic Sea. Considering $H_s$, the comparisons with the above results at a seasonal scale were limited to the analysis by Pomaro et al. [16] in the Adriatic Sea. A rise in the 50th percentile of $H_s$ for all seasons agree with the current findings. A detailed comparison between the present results and those obtained by Pomaro et al. [16] and De Leo et al. [18] is shown in Figure 8. In particular, Figure 8a highlights a comparison between present results and those obtained by Pomaro et a. [16] in terms of annual and seasonal trends of the mean $H_s$ values. A grid point belonging to the northern area of the Adriatic Sea located at a distance of few km to the recording station used by Pomaro et al. [16] was considered. The annual and seasonal mean $H_s$ values obtained by Pomaro et al. [16] show a positive trend as in the current results. With the exception of the spring season, results of this study showed an overall underestimation in comparison with those evaluated by Pomaro et al. [16]. These discrepancies could be due to two reasons. First, the measurement method to evaluate the wave data is different since Pomaro et al. [16] adopted pressure transducers and echo sounders to detect the free surface. Second, the spatial location of the wave nodes is different, since that considered in this analysis proves to be quite close to the coast. Figure 8b describes a comparison between present results and those obtained by De Leo et al. [18] in terms of annual trend of the mean $H_s$ values. To make this comparison, the trend of mean $H_s$ values belonging to the Tyrrhenian, Ligurian, Adriatic, and Ionian Seas were averaged. The current results prove to be generally larger in magnitude if compared with those detected by De Leo et al. [18]. As previously mentioned, a general agreement is limited to the Ionian Sea and, partially, to the Tyrrhenian one, while a different trend behavior occurs for the Ligurian and Adriatic Sea. The differences between these results could be linked to the different space-time resolution of the datasets, i.e., 0.75° × 0.75° and 6 h for the current dataset vs. about 0.1° × 0.1° and 1 h for the dataset used by De Leo et al. [18].

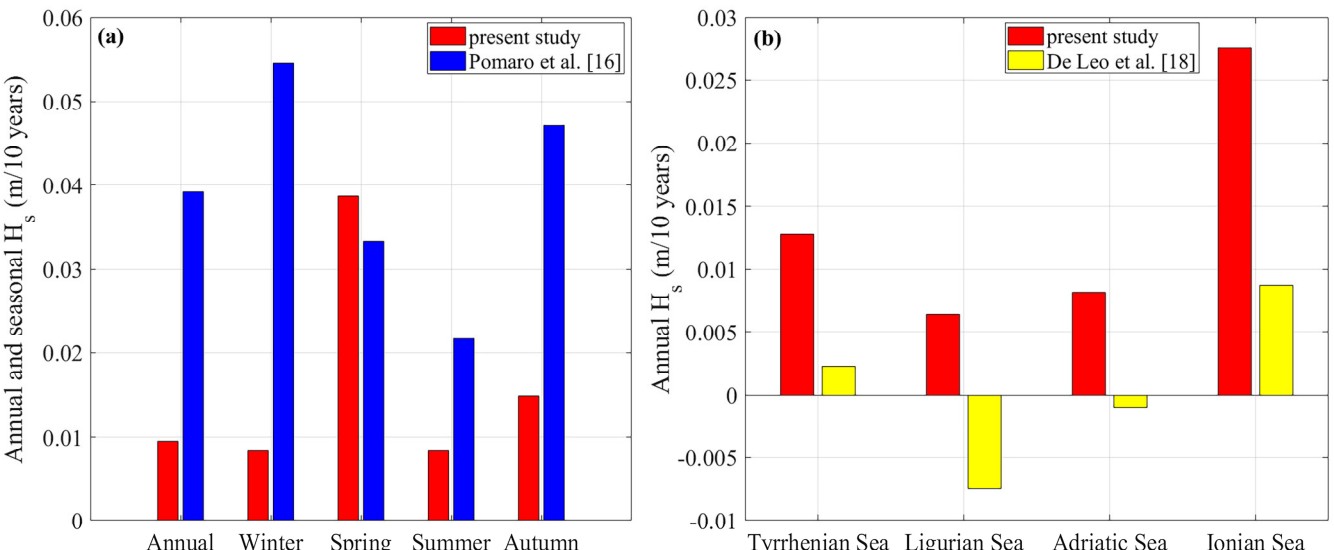

**Figure 8.** (**a**) Annual and seasonal trend of the mean $H_s$ values: comparison between present results and those obtained by Pomaro et al. [16]; (**b**) Annual trend of the mean $H_s$ values: comparison between present results and those obtained by De Leo et a. [18].

## 5. Conclusions

A trend analysis of mean values of significant wave height, energy period, and wave power at annual and seasonal scales was performed considering the Italian seas. To this purpose, the ERA-Interim wave data in the period 1979–2018 were adopted. The Mann–Kendall test was used to identify ongoing trends at yearly and seasonal scales. The magnitude of positive or negative trends was performed through the application of the Theil–Sen estimator.

Regarding the trends of the mean wave parameters at annual scale, the results showed an overall rise of $H_s$, $T_e$, and $P$. This was especially evident off the western area of Sardinia and in the Ionian Sea. The tendency clearly emerges for $T_e$ where all grid points showed a significant positive trend. It was also observed that $H_s$ was the wave parameter having the largest trend variability in the studied seas. The obtained results at a seasonal scale confirmed those identified at a yearly scale. In fact, significant positive trends were detected in the majority of the grid points. The largest increase of $H_s$ was detected in winter and autumn, mainly along the Ionian and Tyrrhenian Seas and off the western part of Sardinia and Corsica. Positive trends of $T_e$ were observed for all seasons, especially during autumn. Regarding $P$, a relevant variability among the seasons was detected, particularly during winter and spring. Highest positive trends of $P$ were mainly evidenced in winter and autumn.

The analyses on the mean values of $H_s$, $T_e$, and $P$ could have a significant impact on various coastal and offshore operations along the Italian seas. In fact, in the future, the potential increase in the mean values could lead to a rise in coastal vulnerability, with a resulting long-term shoreline retreat and a greater occurrence of inundations, as well as problems arising from port and ship operations. At the same time, the positive trend of mean values of $P$ can lead to an improvement of the performances of Wave Energy Converters, particularly for those zones with a strong concentration of wave energy such as the Strait of Sicily and off the western area of Sardinia.

Further investigations will be addressed to detect trends of maximum values of $H_s$, $T_e$, and $P$ along the Italian seas using a wave dataset such as ERA5 by ECMWF characterized by a greater temporal resolution than the present one. Other wave parameters such as wave direction and peak period will be analyzed through the present trend analysis.

**Author Contributions:** The authors contributed equally to this work. All authors have read and agreed to the published version of the manuscript.

**Funding:** This study received no external funding.

**Institutional Review Board Statement:** Not applicable.

**Informed Consent Statement:** Not applicable.

**Data Availability Statement:** The data presented in this study are available on request from the corresponding author.

**Acknowledgments:** The ECMWF wave data were provided by the Meteorological Archival and Retrieval System (MARS) archive with permission of the Italian Air Force. We are indebted to Danilo Algieri Ferraro (University of Calabria) for providing us with the wave data.

**Conflicts of Interest:** The authors declare no conflicts of interest.

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
