# Peer review of "Trend Detection of Wave Parameters along the Italian Seas"

_water, doi:10.3390/w13121634_

Round 1

Reviewer 1 Report

I carefully read the manuscript and, overall, I found it potentially interesting for the readership of Water, since it analyzes the trends over the period 1979-2018 of wave parameters (significant wave height, energy period and wave power) along the Italian Seas by considering wave time series taken from the global atmospheric reanalysis ERA-Interim by ECMWF.

However, I have some major criticisms. In particular:

  • Which is the main novelty of this study compared to already existing studies? How does it advance existing knowledge on the topic? Just the inclusion in the trend analysis of the energy period and wave power cannot be considered enough, in my opinion.
  • In the present form, the discussion of the results is rather poor and, apparently, Authors limit themselves to report the results and describe the figures. I would then encourage the Authors to try to interpret their results, by providing an explanation for observed trends and specific local patterns (e.g. in the Adriatic Sea).      
  • English style and grammar should be largely improved. I list below just few examples, which are not exhaustive.

Specific comments:

  • Section 3.2 could be shortened, since the MK test with Sen’s slope estimator is a standard methodology for calculating trends (you may just mention the references [21-22 and 23], without reporting full details);
  • Figure 2 and similar ones: I would suggest to include in the legend of these figures the symbology for significant / non-significant trends (small vs large squares).
  • English style and grammar:

- there are weird sentences, which need to be rewritten, e.g.: P1.L30-34 (this may be split into two parts); P1.L34-37 (weird construction of the sentence);

- many typos: please proofread the manuscript;

- “less positive values” and similar text: it is clear what do you mean (‘with a smaller trend magnitude’) but the way you wrote it makes no sense.

Reviewer 2 Report

1) Line 75: "Here, the wave parameters Hs and Te were deduced from the global ..."

The method of obtaining wave direction data should also be described in detail.

2) Line 88: "where Sij is the density over the i-th frequency and j-th direction"

Add that Sij is a power spectral density function. 

3) Figure 1 caption: "This is a figure."

"Target sea area and calculation points" are recommended. 

4) equation (7)

Add a sentence that d is the depth of the water. Because the 81st line where d is defined is far away.

5) Lines 160-162: "where the largest positive ones were detected off the western zones of Corsica and Sardinia, along the western part of the Tyrrhenian Sea and within the  Ionian Sea"

The text is redundant. Add a curve surrounding the area to Figure 2b. 
Instead of "Corsica and Sardinia, along the western part of the Tyrrhenian Sea and within the  Ionian Sea ...", it should be describe "the area surrounded by the curve is the area where trend > 0.02". The same method should be applied for Figs 3-7. 

6) figure 3
Add the definition of spring, summer, autumn and winter. Is winter defined from December 1st to the end of February?

7) Lines 310-311 "The present results also agree with the analyses by Pomaro et al. [16]"

Graphs showing this sentence should be added.

8) Lines 315-316: "The ob-tained positive trends on mean  Hs values substantially agreed with those detected by De Leo et al. [18]"

Graphs showing this sentence should be added.

Round 2

Reviewer 1 Report

The Authors have addressed my major comments in a satisfying way and the paper can now be considered acceptable for publication.